# Methodological Standards for Conducting High-Quality Systematic Reviews

**DOI:** 10.3390/biology14080973

**Published:** 2025-08-01

**Authors:** Alessandro De Cassai, Burhan Dost, Serkan Tulgar, Annalisa Boscolo

**Affiliations:** 1Department of Medicine (DIMED), University of Padua, 35127 Padua, Italy; 2Institute of Anesthesia and Intensive Care, University Hospital of Padua, 35127 Padua, Italy; 3Department of Anesthesiology and Reanimation, Ondokuz Mayis University Faculty of Medicine, Samsun 55270, Türkiye; burhandost@hotmail.com; 4Department of Anesthesiology and Reanimation, Samsun University Faculty of Medicine, Samsun 55080, Türkiye; serkantulgar.md@gmail.com; 5Department of Cardiac, Thoracic, Vascular Sciences and Public Health, University of Padua, 35127 Padua, Italy

**Keywords:** systematic review, meta-analysis, evidence synthesis, risk of bias, GRADE, PRISMA, publication bias, narrative synthesis, study selection, guideline

## Abstract

Systematic reviews are an essential way for researchers to collect and evaluate all the scientific studies on a specific topic, helping to make informed decisions in biology policy-making and in other settings. This article explains step-by-step how to plan, conduct and report a high-quality systematic review. It begins with developing a clear research plan, which prevents bias and improves transparency. It then describes how to search for studies in scientific databases, selecting relevant studies, and extracting important information from them. The article also highlights how to assess the quality and reliability of each study to avoid misleading conclusions. Depending on the data, researchers may combine results using statistical methods or summarize them in a clear, descriptive way. Finally, the article emphasizes the importance of assessing how certain we can be about the overall findings. This process ensures that knowledge is based on all available evidence, not just on a few selected studies, and supports better outcomes for society.

## 1. Introduction

Systematic reviews (SRs) are widely regarded as the highest form of evidence within the hierarchy of research designs. However, this position has been increasingly challenged, with some arguing that SRs should be conceptualized not as the apex of evidence-based practice, but rather as a methodological lens through which primary studies are critically appraised, synthesized, and interpreted [1]. From this perspective, SRs serve as structured tools that enable stakeholders to consume, evaluate, and apply the underlying evidence more effectively [1]. This holds true not only within the field of biology but across all areas of scientific research, where systematic reviews play a crucial role in synthesizing evidence and guiding practice, because well conducted SRs minimize bias and offer robust conclusions that inform clinical practice, public health policy, and further research endeavors.

However, not all systematic reviews are of high quality, as the reporting and methodological quality of SRs are inconsistent, although reporting guidelines and quality assessment tools exist [2].

In a survey of 102 SRs and meta-analyses in ecology and evolutionary biology (2010–2019), only ~16% referenced any reporting guideline, and those that did scored significantly higher on reporting quality metrics than the average [3]. Moreover, reviews dating back to the early 2000s highlighted that ecological meta-analyses often lacked essential methodological rigor—such as protocol registration, risk-of-bias assessments, or clear inclusion criteria—revealing a gap relative to established practices in clinical research [4] Poor methodological rigor can result in misleading conclusions, which in turn may negatively impact clinical decision-making and policy formulation. Therefore, understanding the correct methodology is essential not only for researchers but also for reviewers, editors and readers undertaking or interpreting systematic reviews.

Over recent decades, methodological standards for conducting SRs have become increasingly formalized, with the Cochrane Handbook for Systematic Reviews of Interventions widely regarded as the gold standard, particularly for intervention reviews in health care [5]. Although this manuscript addresses SRs across diverse fields and review types—including those beyond clinical interventions—the foundational methods outlined in the Cochrane Handbook remain highly influential and form the methodological bedrock for many review approaches.

This paper seeks to translate these principles into the biological context by addressing unique methodological challenges and incorporating domain-specific tools and resources. The aim of this paper is to offer a comprehensive overview of how to conduct a high-quality systematic review, with detailed discussions of each step based on internationally recognized guidelines and best practices.

## 2. Formulating the Research Question

The first and most fundamental step in a systematic review is to develop a clearly articulated research question [5]. A well-structured question serves as the foundation for the entire review process, guiding search strategies, study selection, and data synthesis. However, even if this is the most crucial step, a study found that 3 out of 10 studies would have needed a major rewording of the research question, as the stated questions were either too vague, inconsistent with the inclusion criteria, or misaligned with the actual objectives and findings [6].

In biology, and more generally in health sciences, the PICO framework and, more frequently, its extended version PICOS, is the most commonly employed tool to structure the research question. PICOS stands for the following: P: Population or patient group, I: Intervention or exposure, C: Comparator or control, O: Outcomes of interest, S: Study design. It has been shown that researchers using the PICO framework have better precision in retrieving relevant studies when querying databases compared to a free keywords strategy [7].

An even more comprehensive variation is PICOTS, even if rarely used, which adds Timeframe (T) to explicitly define the duration over which outcomes are assessed. This is particularly important for ecological or physiological outcomes that vary significantly over time. For instance, if assessing the impact of habitat restoration on bird population density, combining studies measuring outcomes after one breeding season with those measured after five years would likely yield misleading conclusions due to ecological lag effects.

Below three examples of PICOTS questions are presented.

(1)In degraded tropical forest ecosystems, does reforestation with native tree species, compared to natural recovery without intervention, lead to greater increases in species richness and abundance of native fauna, based on long-term field observational studies?Population (P): Degraded tropical forest ecosystems, defined as forests with >50% canopy loss, located within the Amazon Basin (including Brazil, Peru, Colombia), based on WWF ecoregion classifications.Intervention (I): Reforestation efforts involving the planting of native tree species, defined as species naturally occurring in the respective ecoregion, excluding exotics and non-native cultivars.Comparator (C): Natural recovery without active planting or management interventions, defined as passive regrowth following disturbance.Outcome (O): Change in species richness (number of native plant and animal species per hectare, measured via standardized biodiversity surveys) and abundance of native fauna (total counts of individuals per species), assessed separately.Timeframe (T): Minimum of 5 years post-intervention to capture long-term ecological recovery trajectories.Study design (S): Prospective or retrospective observational studies with longitudinal monitoring, excluding short-term experiments (<1 year).(2)In *Pseudomonas aeruginosa* cultures, does exposure to sub-lethal concentrations of ciprofloxacin, compared to no antibiotic treatment, reduce biofilm formation in in vitro experimental studies?Population (P): Clinical isolates of *Pseudomonas aeruginosa* collected from hospitalized adult patients (>18 years) in tertiary care hospitals.Intervention (I): Use of antibiotic A (e.g., ciprofloxacin) administered according to standard dosing guidelines.Comparator (C): Use of antibiotic B (e.g., ceftazidime) or no antibiotic treatment (supportive care only).Outcome (O): Antibiotic resistance development, defined as ≥4-fold increase in minimum inhibitory concentration (MIC) measured by broth microdilution assays.Timeframe(T): Resistance assessed at baseline and after a minimum treatment period of 7 days.Study design (S): Randomized controlled trials or observational cohort studies with appropriate resistance-testing protocols.(3)In *Arabidopsis thaliana*, how does exposure to salicylic acid, compared to untreated controls, influence the expression of pathogen-resistance genes, based on transcriptomic data from time-course experiments?Population (P): Laboratory-grown *Arabidopsis thaliana* ecotype Col-0 plants at the rosette stage (4 weeks post-germination), cultivated under standardized photoperiod and temperature conditions.Intervention (I): Treatment with 1 mM salicylic acid (SA), applied via foliar spray, simulating immune signaling induction.Comparator (C): Untreated control plants sprayed with distilled water under identical conditions.Outcome (O): Differential expression levels of key defense-related genes (e.g., *PR1*, *NPR1*, *WRKY70*) quantified using RNA-seq and normalized counts (TPM/RPKM), with validation by qRT-PCR.Timeframe (T): Gene expression measured at baseline (0 h), 1 h, 6 h, and 24 h post-treatment to capture temporal transcriptional dynamics.Study Design (S): Controlled in vitro experiments with randomized assignment and technical triplicates, using at least two independent biological replicates per time point.

While PICO is a strong starting point for many clinical and intervention-based studies, exploring these alternatives can lead to a more focused and effective research question for other contexts. For example, “SPIDER” (sample, phenomenon of interest, design, evaluation, research type), is designed specifically to identify relevant qualitative and mixed-method studies [8]; however, its use is more widely limited than the PICOS framework [9] and is exemplified as follows:(4)How do adult members of Indigenous communities living adjacent to protected wildlife reserves in the Amazon Basin perceive the social and ecological impacts of community-based conservation programs, as explored through in-depth qualitative interviews? S (Sample): Adult members (≥18 years old) of Indigenous communities residing within 10 km of protected wildlife reserves in Brazil’s Amazon Basin (e.g., Kayapó, Yanomami territories); PI (Phenomenon of Interest): Perceptions and lived experiences related to community-based conservation programs, including views on biodiversity, land use, and cultural autonomy; D (Design): Qualitative studies using semi-structured or in-depth interviews, ethnographic methods, or participatory focus groups; E (Evaluation): Thematic data describing perceived benefits (e.g., increased wildlife), concerns (e.g., reduced hunting access), and trust in conservation agencies; R (Research type): Qualitative studies only (excluding quantitative surveys or mixed-methods designs unless qualitative data are reported separately and in detail).(5)How do early-career molecular biology researchers perceive the use of CRISPR-Cas9 gene editing technologies in basic versus translational research contexts, as explored through qualitative interviews? S (Sample): Early-career researchers (PhD students and postdoctoral fellows with ≤5 years of experience) working in molecular biology laboratories across research institutions in the United States and Europe; PI (Phenomenon of Interest): Perceptions, ethical concerns, and decision-making processes related to the use of CRISPR-Cas9 gene editing in both basic research (e.g., model organism studies) and translational/clinical applications (e.g., gene therapy); D (Design): Qualitative studies using semi-structured interviews, online focus groups, or ethnographic observation in lab settings; E (Evaluation): Thematic data describing perceived benefits (e.g., research acceleration, therapeutic potential), risks (e.g., off-target effects, ethical misuse), and institutional pressures (e.g., funding priorities, publication expectations); R (Research type): Qualitative studies only, excluding mixed-method studies unless the qualitative data are clearly reported and analyzed separately.

Several alternatives to the PICOS and SPIDER framework exist for the structuring of research questions, especially for qualitative and mixed-method studies [10]. SPICE (Setting, Perspective, Intervention, Comparison, Evaluation) focuses on the context and is useful in social research—for example, assessing how farmers in arid regions perceive agroforestry compared to monoculture. PEO (Population, Exposure, Outcome) simplifies qualitative inquiries, e.g., exploring the experiences of fishers affected by marine protected areas. For policy and service evaluation, ECLIPSE (Expectation, Client group, Location, Impact, Professionals, Service) frames questions around service improvement, such as evaluating community expectations of local conservation programs. CIMO (Context, Intervention, Mechanism, Outcome) suits organizational or environmental management research, e.g., examining how community-led wetland restoration (intervention) in urban areas fosters stewardship (mechanism) and improves biodiversity (outcome).

## 3. Developing a Protocol

Developing and pre-registering a protocol serves two key purposes [11]. First, it provides structured guidance for each stage of the review, ensuring that all team members understand the workflow and process. Second, it acts as a safeguard against methodological bias by promoting transparency and accountability. Several platforms are available for protocol pre-registration. The most widely recognized is PROSPERO [12], a public registry specifically dedicated for systematic reviews. Another widely used option is the Open Science Framework (OSF), which supports the pre-registration of a broader range of research projects beyond systematic reviews. Additionally, researchers may consider publishing their protocol as an article in a peer-reviewed journal that accepts protocol submissions, which provides the added benefit of formal peer review and greater visibility.

Some registries guide researchers by requiring them to complete structured fields that cover the key aspects of the systematic review. Others allow researchers to simply upload a study protocol without predefined fields. The first approach is better suited to inexperienced systematic review authors, as it reduces the risk of producing an incomplete protocol that may omit essential methodological details. Regardless of the format, certain critical information should always be provided: the objectives and research questions, eligibility criteria, information sources (such as databases and registries), detailed search strategies, the study selection process, the data extraction plan, strategies for assessing risk of bias, and methods for data synthesis—including plans for meta-analysis or narrative synthesis. Additionally, any intended subgroup or sensitivity analyses should be specified if applicable.

In relation to minimizing the risk of bias, it is important to emphasize that a poorly detailed or vague protocol carries similar risks to conducting a systematic review without a protocol at all [13] and pre-registering the protocol helps reduce the risk of selective inclusion of studies or outcome data, which could otherwise lead to overestimation or underestimation of the systematic review’s results [14]. Inadequate protocol transparency may lead readers to underestimate the potential for bias, thereby limiting the critical appraisal of the review’s methodological rigor.

## 4. Formulating the Search Strategy

A comprehensive search strategy aims to capture all relevant studies [15]. While the PICOS model is fundamental in formulating the research question, it should not be directly applied in full when designing the search strategy. In fact, evidence suggests that incorporating all elements of PICOS—particularly the comparator and outcomes—can reduce the accuracy of search results, as these elements are often inconsistently reported in titles and abstracts [16], and current guidelines generally recommend that search strategies focus on the population, intervention(s), and study design [17].

The Cochrane Handbook for Systematic Reviews of Interventions [5] advises searching multiple databases to minimize the risk of missing relevant studies. Commonly used databases include MEDLINE/PubMed, EMBASE, Scopus, Cochrane Central Register of Controlled Trials (CENTRAL), CINAHL, PsycINFO, CABI and BIOSIS. However, this list is not exhaustive, and researchers are encouraged to expand their search to as many relevant databases as resources allow. In addition to these general databases, it is essential to highlight biology domain-specific resources, such as GenBank for genetic sequences, BOLD Systems for DNA barcoding data, and GBIF for biodiversity records. Furthermore, repositories, like Dryad, BioStudies, and Zenodo, facilitate open data sharing and promote transparency and reproducibility in biological research. Utilizing these resources helps researchers access diverse datasets, essential for addressing complex biological questions across molecular biology, ecology, and evolutionary studies.

It is crucial for researchers to recognize that each database has its own controlled vocabulary and search syntax. While a detailed explanation of database-specific search languages falls outside the scope of this article, search strings typically combine database specific controlled vocabulary (e.g., MeSH terms) with free-text keywords. Boolean operators (AND, OR, NOT) and truncation symbols are employed to refine searches. Furthermore, to ensure transparency and reproducibility, the exact search strategy used for each database—including the date the search was conducted—should be reported, typically as supplementary material.

Additional strategies should be employed to maximize the retrieval of relevant articles for a systematic review. These include searching the grey literature, such as theses, dissertations, conference proceedings, and clinical trial registries (e.g., ClinicalTrials.gov, WHO ICTRP). Citation tracking—both backward (screening the reference lists of included studies) and forward (identifying articles that have cited the included studies, often referred to as snowballing)—is another effective approach. Furthermore, contacting the authors of included studies can help identify unpublished or ongoing studies that may otherwise be missed.

Other recommended strategies include searching preprint servers, consulting subject-specific repositories, and exploring regulatory agency reports or industry submissions, where applicable. All these supplementary search methods should be clearly outlined in the pre-registered protocol.

Another interesting aspect that warrants discussion is the use of AI-driven platforms to enhance the efficiency and accuracy of the systematic review process [18]. These technologies can assist with tasks such as formulating the search strategy, de-duplication, screening, data extraction, and even risk-of-bias assessment. When used appropriately, these tools can streamline workflows, reduce reviewer burden, and improve consistency—though human oversight remains essential to ensure methodological rigor. However, despite ongoing research and promising developments, there remains uncertainty regarding the full scope of their future applications and limitations. Moreover, the use of AI in systematic reviews raises important ethical concerns, including transparency, accountability, and the potential for bias embedded in algorithmic decision-making.

## 5. Study Screening and Selection

Study screening is typically conducted in two stages: (1) title and abstract screening, and (2) full-text screening. At each stage, screening decisions should be based on predefined inclusion and exclusion criteria, which are developed from the review’s PICOTS (or equivalent) framework, but are not identical to it.

While PICOTS helps structure the research question, inclusion and exclusion criteria must translate that question into concrete, testable conditions. For example, if the “population” in PICOTS is “tropical forest ecosystems with >50% canopy loss,” the inclusion criteria must further specify what qualifies as “tropical,” and which geographic regions are eligible. Similarly, if the study design is limited to “observational studies with ≥5 years of follow-up,” the exclusion criteria must rule out short-term experiments, cross-sectional studies, or studies without clear follow-up durations.

During title and abstract screening, reviewers should generally err on the side of inclusion—retaining any study that potentially meets the criteria. This stage is intended to be sensitive, rather than specific. Full-text screening, in contrast, is more stringent and is used to exclude studies that do not fully meet the eligibility criteria. This step requires careful judgment and often necessitates pilot screening to calibrate reviewer decisions.

To minimize the risk of bias, this process should be performed independently by at least two reviewers, with disagreements resolved by consensus or by a third reviewer. Emphasizing a double-screening approach is crucial, as evidence indicates that single-reviewer screening is associated with the omission of a significant number of eligible studies [19].

However, manually screening hundreds—or even thousands—of records is highly resource-intensive and, in some cases, may become impractical. To address this challenge, various web-based screening tools have been developed to improve efficiency and manageability. Popular examples include Abstrackr [20], Rayyan [21], Covidence [22], and EPPI-Reviewer [23], which facilitate collaborative screening, streamline decision tracking, and reduce reviewer workload without compromising methodological rigor.

The overall results of the screening and selection process are typically summarized using the PRISMA flow diagram (Figure 1), which visually documents each step of the process.

To improve the overall quality, researchers are encouraged to provide, as supplementary material, a dataset listing all studies retrieved, screened, included, or excluded, along with detailed reasons for exclusion at each stage of the selection process.

## 6. Data Extraction

Data extraction is the systematic collection of information from included studies [24]. This step should be guided by a predefined data extraction form, often piloted on a subset of studies to ensure clarity and consistency. Essential data typically extracted include study identifiers (authors, year, journal), study design (e.g., RCT, cohort, case-control), population characteristics (e.g., sample size, demographics), intervention and comparator details, outcomes measured and timepoints, results (e.g., effect sizes, confidence intervals), funding sources, and potential conflicts of interest Similarly to the study screening and selection process, data extraction should be performed independently by at least two researchers, as evidence indicates that relying on a single extractor can lead to 21% more errors compared to a double-extraction approach [25]. Although this method nearly doubles the time required compared to a single-extractor approach, the substantial improvement in accuracy and reduction in errors justifies the additional effort [25]. Readers should be aware that various tools, such as EPPI-Reviewer, Covidence, and Excel-based templates, can support data extraction.

## 7. Assessing Risk of Bias

Evaluating the methodological quality of included studies is a cornerstone of a rigorous systematic review, as it directly influences the reliability and validity of the review’s conclusions. Several standardized tools have been developed to assess the risk of bias in different types of studies (Table 1) [26,27,28,29,30,31,32,33,34,35,36,37,38,39,40,41].

For randomized controlled trials (RCTs), the most widely recommended tools are the Cochrane Risk-of-Bias 2 (RoB 2) tool [27] and the Jadad scale [28], with RoB 2 offering a more comprehensive, domain-based assessment. For non-randomized studies of interventions, tools such as ROBINS-I (Risk Of Bias In Non-randomized Studies of Interventions) [29] and ROBINS-E (for exposures) [30] are commonly used, alongside the older but still widely applied Newcastle–Ottawa Scale (NOS) [31].

For diagnostic accuracy studies, the QUADAS-2 tool is the standard [32]. When dealing with qualitative studies, researchers frequently employ the CASP (Critical Appraisal Skills Programme) checklists [33]. For preclinical animal studies, the SYRCLE Risk-of-Bias tool is specifically designed to capture methodological concerns relevant to laboratory in vivo-based research [34].

Each tool is structured around a set of domains that reflect potential sources of bias relevant to the study design. While the specific domains vary between tools, they commonly assess areas such as the following:(1)Selection bias (e.g., randomization, allocation concealment)(2)Performance bias (e.g., blinding of participants and personnel)(3)Detection bias (e.g., blinding of outcome assessors)(4)Attrition bias (e.g., incomplete outcome data)(5)Reporting bias (e.g., selective outcome reporting)

The assessment of each domain typically involves assigning a judgment categorized as low risk, high risk, or unclear risk of bias, often visually represented by colors (e.g., green for low risk, yellow for unclear, and red for high risk). Some tools, such as RoB 2 and ROBINS-I, generate an overall risk-of-bias judgment for each study based on domain-level assessments, while others, like the Newcastle–Ottawa Scale, provide a score without a formal overall risk category. The results of risk-of-bias assessments are commonly presented in summary tables or graphical displays, such as traffic light plots or weighted bar charts. However, to enhance transparency and allow readers to fully understand how judgments were made, researchers should also provide detailed justifications for each risk-of-bias judgment assigned to each domain for every included study. Ideally, these justifications should be made available in the main text, appendices, or as supplementary material. The process should be conducted independently by at least two reviewers to minimize subjective judgment and ensure reliability.

## 8. Data Synthesis–Quantitative Synthesis (Meta-Analysis)

When a group of studies addressing the same research question are sufficiently similar in terms of participants, interventions, comparators, and outcomes, it is often appropriate to conduct a meta-analysis. Meta-analysis is a statistical technique that combines the results of individual studies to produce a single pooled estimate of the effect size, thereby increasing the overall statistical power and providing a more precise estimate than any single study alone. This approach also allows researchers to explore variability among study results and can offer insights into patterns that may not be apparent from individual studies.

A fundamental aspect of meta-analysis is the choice of the appropriate effect measure. Commonly used effect measures include the risk ratio (RR), odds ratio (OR), and hazard ratio (HR) for dichotomous outcomes, while continuous outcomes are often summarized using the mean difference (MD) or standardized mean difference when different scales are used. Selecting the correct effect measure depends on the nature of the outcome data and the clinical question. Another critical consideration is the degree of heterogeneity among the included studies. Heterogeneity refers to the variation in study outcomes beyond what would be expected by chance alone. It can arise from differences in study populations, interventions, outcome definitions, or study quality. The most commonly used statistic to quantify heterogeneity is the I^2^ statistic, which represents the percentage of total variation across studies due to heterogeneity rather than chance. According to guidelines proposed by Higgins and colleagues (2003) [41], I^2^ values of approximately 0–40% might not be important, 30–60% may represent moderate heterogeneity, 50–90% may represent substantial heterogeneity, and 75–100% considerable heterogeneity. The importance of the observed value of I^2^ depends on the magnitude and direction of effect and strength of evidence of heterogeneity (for example *p*-value from the chi square test); however, the interpretation of heterogeneity is complex but important and should always be explored whenever possible (even through sensitivity analysis). It is important to remember that the uncertainty in the I^2^ is substantial when the number of studies is small.

When potential sources of heterogeneity are known or suspected—such as differences in populations, interventions, or study quality—sensitivity analyses should be conducted. These involve re-running the meta-analysis after excluding specific subsets of studies (e.g., low-quality studies, studies with extreme effect sizes) to assess how robust the overall findings are, and they are helpful in potentially identifying the source of heterogeneity.

Regarding model choice, two primary approaches exist: fixed-effect and random-effects models. Fixed-effect models operate under the assumption that all included studies estimate the same underlying effect size, and observed differences are due solely to sampling error. In contrast, random-effects models acknowledge that true effect sizes may vary across studies due to genuine clinical or methodological differences and therefore incorporate this variability into the pooled estimate. The choice between a fixed or a random effect model should be performed a priori, and a fixed effects models should be preferred over random effect models only if researchers are absolutely certain that all the studies included do not have confounding effects or heterogeneity, a situation that rarely takes place.

Finally, assessing publication bias is a crucial step in meta-analysis to determine whether the included studies disproportionately represent positive or statistically significant findings, which can lead to an overestimation of the true effect. One common method for detecting publication bias is the funnel plot [42], a scatterplot that displays individual study effect sizes against a measure of their precision, typically the standard error or sample size. The underlying principle of funnel plots is that smaller studies, which have less precise estimates, should show a wider spread of effect sizes, while larger studies with greater precision tend to cluster closer to the true effect. In the absence of bias, the plot should resemble a symmetrical inverted funnel.

When there are a sufficient number of studies (usually ten or ore) publication bias can be evaluated with funnel plots through visual inspection for asymmetry, potentially suggesting the presence of publication bias or other small-study effects. However, visual interpretation can be subjective and inconsistent. Therefore, statistical tests such as Egger’s regression test can be employed to provide a more objective and reproducible assessment of funnel plot asymmetry [43].

In addition to traditional pairwise meta-analyses, which compare two interventions at a time, network meta-analyses (NMAs) allow for simultaneous comparisons of multiple interventions, even if some have not been directly compared in any individual study. NMA integrates both direct and indirect evidence using a common comparator (e.g., placebo), enabling the ranking of interventions in terms of effectiveness. This approach is particularly valuable in fields where multiple treatment or management strategies exist, and head-to-head trials are sparse.

## 9. Data Synthesis–Qualitative Synthesis (Narrative Synthesis)

When conducting a systematic review, a meta-analysis may not be feasible or appropriate when there is substantial heterogeneity across studies—whether in terms of interventions, populations, outcome measures, or study designs—or when the available data are insufficient for meaningful quantitative synthesis, particularly when key outcomes are reported in too few studies or with inconsistent measures. In such cases, researchers should perform a narrative synthesis, which involves systematically summarizing and explaining the findings without statistically combining them. This approach typically organizes results thematically, categorically, or according to key characteristics, such as intervention type, population group, or outcome domain.

To improve the rigor and transparency of narrative syntheses, the SWiM (Synthesis Without Meta-analysis) guideline was developed [44]. SWiM provides structured recommendations on how to transparently report the methods used for grouping studies, how findings are synthesized, and how certainty in the evidence is assessed. In particular, key components of SWiM include detailed descriptions of how studies were grouped or clustered for synthesis, the criteria and rationale for combining or comparing results, and the approach used to assess the certainty of evidence. SWiM also emphasizes the importance of reporting how data were extracted and how any quantitative transformations were conducted when applicable, such as when effect sizes are calculated but not pooled.

Another important aspect of SWiM is promoting clarity in presenting results, encouraging authors to use tables and figures effectively to summarize study characteristics and findings. This helps readers understand the evidence without relying solely on textual descriptions, which can sometimes be ambiguous or incomplete. The guideline also advocates for explicit discussion of the limitations of the narrative synthesis approach, including potential biases and challenges arising from lack of statistical pooling.

## 10. Assessing Certainty of Evidence

One of the final and most critical steps in conducting a systematic review is the assessment of the certainty of evidence, which reflects the degree of confidence that the estimated effect is close to the true effect. The GRADE (Grading of Recommendations, Assessment, Development and Evaluation) approach is the internationally recognized standard for evaluating the certainty of evidence [45] and should be applied in systematic reviews regardless of whether a meta-analysis is performed or not. GRADE provides a structured and transparent framework to assess the strength of evidence based on key domains, ensuring that the conclusions of the review are well-founded, even when quantitative synthesis is not feasible [44].

GRADE assesses certainty based on five key domains. The first is risk of bias, which considers whether limitations in study design or execution may have introduced systematic errors. The second is inconsistency, which refers to unexplained heterogeneity or variability in results across studies. The third domain, indirectness, evaluates the applicability of the evidence to the research question, particularly when there are differences in populations, interventions, or outcomes. The fourth domain is imprecision, which addresses whether the evidence is weakened by wide confidence intervals or small sample sizes that reduce the reliability of the effect estimate. Finally, publication bias assesses whether the available evidence might be skewed due to the selective publication of studies with positive or significant findings.

Based on an assessment of these domains, the certainty of evidence is classified into one of four levels: high, moderate, low, or very low. High-certainty evidence indicates strong confidence that the true effect lies close to the estimate, while very low certainty reflects considerable uncertainty about the effect estimate. The results of the GRADE assessment are commonly presented in a Summary of Findings (SoF) table, which succinctly displays the main outcomes, effect sizes, confidence intervals, and corresponding GRADE ratings.

## 11. Alternative Approaches to Evidence Synthesis

While this manuscript focuses on broadly applicable, quantitative synthesis approaches (e.g., GRADE, SWiM, Cochrane Handbook), it is important to acknowledge alternative methodologies developed for different types of evidence and research questions. For instance, the RAMESES (Realist And MEta-narrative Evidence Syntheses: Evolving Standards) [46] framework supports realist reviews, which aim to understand how and why interventions work (or fail) in particular contexts—especially in complex, system-level settings. Similarly, in qualitative research, meta-aggregation (as used in the Joanna Briggs Institute model) [47] offers a structured approach for synthesizing findings while preserving participants’ original meaning. Unlike thematic synthesis, meta-aggregation emphasizes transparency and alignment with primary study conclusions. These alternative methodologies serve distinct purposes and are complementary to the general frameworks described here. Researchers should carefully consider their review goals and the nature of the available evidence when selecting an appropriate synthesis method.

## 12. Unique Methodological Challenges in Biological Research

Biological research, particularly in ecology, evolutionary biology, and related subfields, faces several methodological challenges that are distinct from clinical or health-related studies. These challenges stem from the complexity of living systems and their interactions with highly variable environments.

 **(a)**
**Non-standardized outcome measures in ecological and evolutionary studies**


Unlike clinical trials with well-established and standardized endpoints (e.g., blood pressure or mortality), ecological and evolutionary studies often rely on diverse outcome measures, such as species richness, biomass, gene expression levels, or behavioral traits. This diversity can make it difficult to directly compare results across studies or synthesize findings through meta-analyses [48]. Moreover, these outcomes may be measured using different methodologies, scales, or temporal resolutions, increasing methodological heterogeneity and complicating systematic reviews [49].

 **(b)**
**Species-specific or strain-specific biological responses**


Variation between species, strains, or populations is a hallmark of biological research, reflecting differences in genetics, physiology, life history, and ecological niches. Such biological heterogeneity means that responses to experimental treatments or environmental factors can differ markedly, even within seemingly similar contexts. For example, a pesticide might have a strong toxic effect on one insect species but be benign to another. Failing to account for these differences risks overgeneralizing results and undermines the validity of pooled analyses [50]. Researchers must carefully consider taxonomic and genetic diversity when designing studies and synthesizing data.

 **(c)**
**High heterogeneity due to environmental contexts in biodiversity or phylogenetic research**


Environmental variability introduces another significant source of heterogeneity in biological research. Spatial factors (e.g., latitude, habitat type), temporal dynamics (e.g., seasonal variation), and anthropogenic impacts (e.g., pollution, land use changes) all affect biological outcomes and can lead to conflicting or context-dependent findings. Biodiversity studies, in particular, must grapple with this complexity when synthesizing data from multiple ecosystems or geographic regions. Phylogenetic relationships further complicate analyses, as closely related species may share traits that influence responses, requiring sophisticated statistical models to control for evolutionary relatedness [51,52]. These challenges highlight the need for explicit modeling of heterogeneity and sensitivity analyses in biological meta-research.

## 13. Reporting the Systematic Review

Once the SR has been completed, the next crucial step is to report it in a clear, transparent, and comprehensive manner. Fortunately, researchers can rely on the PRISMA 2020 reporting guideline (Preferred Reporting Items for Systematic Reviews and Meta-Analyses) [53], which provides a structured framework to ensure that all essential components of the review are adequately reported. Adhering strictly to PRISMA enhances both the transparency and reproducibility of the review process. The PRISMA 2020 checklist consists of 27 items covering all sections of a systematic review, from title and abstract to methods, results, and discussion. Researchers are strongly encouraged to carefully read the checklist before drafting the manuscript and to complete it thoroughly prior to submission to a peer-reviewed journal. It is highly recommended that researchers annotate the checklist not only with the corresponding page and line numbers but also by including the exact sentences or sections that address each item. This practice greatly facilitates the work of editors, peer reviewers, and other research teams in verifying compliance with reporting standards, further strengthening the credibility, clarity, and transparency of the systematic review.

## 14. Conclusions

The responsibility of conducting a high-quality systematic review lies firmly with researchers. Beyond simply aggregating studies [54], researchers must engage in several steps that have to be planned at the very beginning of the development of a meta-analysis, as we have tried to illustrate in our review. Accurate planning, rigorous methodology and transparent reporting are the key-components in producing evidence that is both credible and useful. High-quality systematic reviews are foundational to evidence-based medicine, informing clinical decision-making, shaping community policies, and guiding future research priorities. Without strict adherence to internationally recognized best practices, systematic reviews risk bias, irreproducibility and, ultimately, misleading conclusions that could affect patient care and resource allocation. Several tools have been developed to help researchers in this hard work, such as the PRISMA 2020 guideline, the Cochrane Handbook for Systematic Reviews of Interventions, and the GRADE approach enhances the methodological rigor and transparency of reviews. PRISMA ensures comprehensive and standardized reporting, making the review accessible and interpretable to a wide audience, including clinicians, policymakers, and fellow researchers. The Cochrane Handbook offers detailed guidance on every stage of the review process, from framing the research question and designing search strategies to assessing risk of bias and synthesizing evidence. Meanwhile, GRADE provides a systematic method to evaluate the certainty of evidence, helping stakeholders understand how confident they can be in their findings.

Moreover, researchers have an ethical obligation to minimize bias and maximize transparency throughout the review. This includes pre-registering protocols, conducting thorough and unbiased literature searches, independently screening and extracting data, and openly sharing data and analytic methods. Such transparency facilitates reproducibility and critical appraisal, strengthening trust in the findings. Additionally, acknowledging and addressing limitations—whether in study quality, heterogeneity, or publication bias—demonstrates intellectual honesty and allows readers to interpret results within the appropriate context.

Ultimately, high-quality systematic reviews serve as powerful tools that synthesize vast amounts of information into actionable knowledge. By committing to excellence in their conduct, researchers ensure that these reviews remain invaluable resources that advance science and inform policies that affect lives worldwide.

In summary, this guide aims to consolidate widely accepted standards for conducting systematic reviews and adapting them to the specific needs of researchers in the biological sciences. From diverse study designs and outcome measures to field-specific challenges, like ecological variability and non-standardized interventions, systematic reviews in biology require careful methodological attention. By contextualizing established frameworks—such as PRISMA, GRADE, and SWiM—and highlighting additional tools relevant to biology (e.g., SPIDER), this manuscript supports researchers in producing high-quality, transparent, and reproducible evidence syntheses. Strengthening the methodological foundation of systematic reviews in biology is essential for advancing reliable, evidence-based decision-making in conservation, ecology, agriculture, and biomedical science.

## Figures and Tables

**Figure 1 biology-14-00973-f001:**
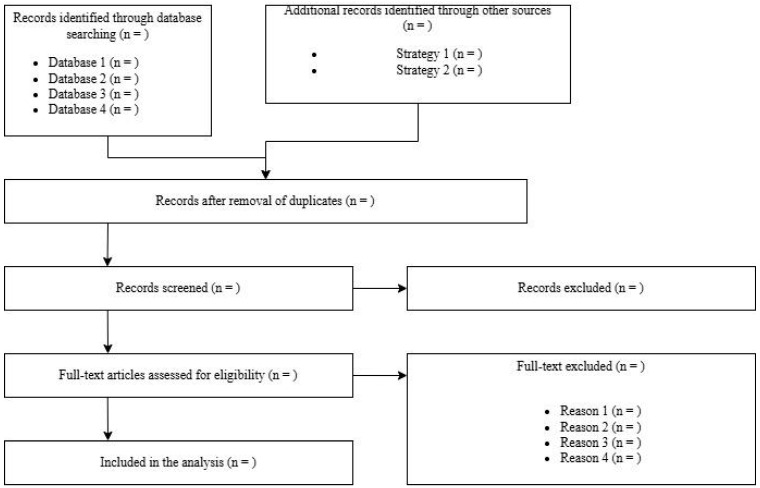
Illustrative example of a PRISMA flowchart.

**Table 1 biology-14-00973-t001:** Risk-of-bias tools.

Risk-of-Bias Tool	Study Type
Cochrane Risk of Bias 2 (RoB 2) [27]	Randomized Controlled Trials
Jadad Scale [28]	Randomized Controlled Trials
ROBINS-I (Risk Of Bias In Non-randomized Studies—of Interventions) [29]	Non-Randomized Intervention Studies
ROBINS-E (Risk Of Bias In Non-randomized Studies—of Exposures) [30]	Observational Exposure Studies (Cohort, Case-Control)
Newcastle-Ottawa Scale (NOS) [31]	Cohort and Case-Control Studies
QUADAS-2 (Quality Assessment of Diagnostic Accuracy Studies) [32]	Diagnostic Accuracy Studies
CASP (Critical Appraisal Skills Programme) [33]	Qualitative Studies
SYRCLE Risk-of-Bias Tool [34]	Animal Intervention Studies
AMSTAR 2 (A Measurement Tool to Assess Systematic Reviews) [35]	Systematic Reviews and Meta-Analyses
MINORS (Methodological Index for Non-Randomized Studies) [36]	Non-Randomized Surgical Studies
Effective Public Health Practice Project (EPHPP) Quality Assessment Tool [37]	Public Health Intervention Studies
Joanna Briggs Institute (JBI) Critical Appraisal Tools [38]	Various Study Designs
ROBIS (Risk of Bias in Systematic Reviews) [39]	Systematic Reviews
AXIS Tool [40]	Cross-Sectional Studies

## Data Availability

All data referenced in the manuscript are fully available within the manuscript itself.

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
