# Peer review of "Methodological Standards for Conducting High-Quality Systematic Reviews"

_biology, 2025, doi:10.3390/biology14080973_

Round 1
Reviewer 1 Report
Comments and Suggestions for Authors
-
In the Abstract, consider simplifying complex sentences (e.g., lines 35–37) to improve readability for non-native English speakers.
-
In Section 4 (Search Strategy), you might briefly comment on the use of automation tools or AI-driven platforms increasingly adopted in SRs.
Author Response
Q1:In the Abstract, consider simplifying complex sentences (e.g., lines 35–37) to improve readability for non-native English speakers.
A1:We modified the sentence as follows: "Both meta-analysis (quantitative approach) narrative synthesis (qualitative approach) are discussed in detail"
Q2: In Section 4 (Search Strategy), you might briefly comment on the use of automation tools or AI-driven platforms increasingly adopted in SRs.
A2: Added the following paragraph :"
Another interesting aspect that warrants discussion is the use of AI-driven platforms to enhance the efficiency and accuracy of the systematic review process [14]. These technologies can assist with tasks such as formulating the search strategy, de-duplication, screening, data extraction, and even risk-of-bias assessment. When used appropriately, these tools can streamline workflows, reduce reviewer burden, and improve consistency—though human oversight remains essential to ensure methodological rigor. However, despite ongoing research and promising developments, there remains uncertainty regarding the full scope of their future applications and limitations. Moreover, the use of AI in systematic reviews raises important ethical concerns, including transparency, accountability, and the potential for bias embedded in algorithmic decision-making."
Reviewer 2 Report
Comments and Suggestions for Authors
De Cassai and coworkers have written a summary of best-practice methods for conducting a high-quality systematic review. Given the number of very poor-quality reviews that are published every year, such a guide would be useful. Generally speaking this article is pretty good. However, the manuscript requires some changes in order to make it reflect actual best practice. This should, however, be more than possible.
Major points
- Your aim here was clearly to give a relatively brief overview of the most important aspects of doing a good quality SR. That is absolutely fine. However, as you are no doubt well aware, every one of your sections could be (and probably is) an entire book. Because it is not possible to cover all these details in your overview, you should provide the reader with a citation for gold standard, detailed instructions and guidance for each of the topics you cover.
- Introduction: the gold standard for conducting systematic reviews of interventions is of course the Cochrane Collaboration. I understand that your manuscript considers systematic reviews of fields other than medicine and types other than interventions, but the Cochrane Handbook really is the bible for intervention reviews and should be discussed and cited.
- Lines 70-71: I am not sure what you mean by “would have needed a major rewording of the research question”. You should expand a bit on this (one sentence). For example, perhaps you mean: “…would have needed a major rewording of the research question. That is, the findings of the study did not align with the research question”. Whatever it is that you mean, please add an explanation in your text.
- Lines 72-76: you have forgotten the most comprehensive version: PICOTS – this includes the timeframe, which is vital for ensuring the data in the studies are comparable. I know some people include this in the Study design part, but in my experience, most authors do not remember to include the time component in their PICOS. Imagine you have an outcome like HbA1c, which is a measure of long-term glucose control, and then they measured the effects of an intervention after a day, and you combined this study with studies at 2 weeks and 6 months and 9 years. This would be inappropriate and inaccurate. Please include a short discussion of time in this section.
- Lines 79-96: these are actually not very comprehensive PICOS questions. They leave a lot of room for a study author to change the inclusion of studies to suit a predetermined preferred outcome. If I got a protocol to review with PICOS like this, I would send the authors back to fix them.
For the forest ecosystem question, it should be more like:
In degraded (defined as …) tropical forest ecosystems [anywhere they exist? Only in certain countries? What definition of tropical?], does reforestation (defined as…) with native tree species [what does “native” mean here? Native to the country, to the region, to that forest?], compared to natural recovery without intervention (defined as…), lead to greater increases species richness (defined as…) and abundance of native fauna [abundance = sheer numbers, increased numbers per species, something else? Also native meaning what here?] based on long-term [6 months, 1 year, 100 years? What is the time frame here?] observational studies [of any kind? Prospective only?]?
Similarly, the Pseudomonas question is not nearly detailed enough and should not be presented as a model for a good quality PICOS. The same is true for your example of a SPIDER question too.
You need to rewrite these to reflect best practice. - Search strategy: you mention (lines 177-178) that clinical trial registries “are commonly employed” however, at least for reviews of interventions, this is absolutely essential. The Cochrane Collaboration considers that authors should at LEAST search the ICTRP. I have found on many occasions that data are available in, for example, ClinicalTrials.gov, that are not published and will never be published. This is frequently because the study did not find what the study sponsor would like. Neglecting to include clinical trials is a major risk of bias. You should change your manuscript to reflect this.
- Study screening: you list all the sections of a PRISMA flow diagram (which is unnecessary because the reader can see this themselves), but you spend very little space talking about the purpose and details of screening.
You need to include a section on the creation of inclusion and exclusion criteria from the PICOTS, because it is NOT the same thing. You should also talk about the difference between screening abstracts (erring on the side of inclusion for example) and screening full texts (whittling down to truly relevant studies). - Heterogeneity (lines 284-291): although you write “approximately” when providing cut-offs for low moderate, and high heterogeneity, this is NOT what the Cochrane Handbook states. Heterogeneity is complex and nuanced, and the use of simple cutoffs is not appropriate. The Cochrane Handbook ACTUALLY says:
“Thresholds for the interpretation of the I2 statistic can be misleading, since the importance of inconsistency depends on several factors. A rough guide to interpretation in the context of meta-analyses of randomized trials is as follows:- 0% to 40%: might not be important;
- 30% to 60%: may represent moderate heterogeneity*;
- 50% to 90%: may represent substantial heterogeneity*;
- 75% to 100%: considerable heterogeneity*.
*The importance of the observed value of I2 depends on (1) magnitude and direction of effects, and (2) strength of evidence for heterogeneity (e.g. P value from the Chi2 test, or a confidence interval for I2: uncertainty in the value of I2 is substantial when the number of studies is small).”
This is in section 10.10.2 of the current Cochrane Handbook.
- Model choice: you nearly got this right. Your description of fixed vs random effects models is correct, but it is wrong to state that a random effects model is chosen as a consequence of substantial heterogeneity (even though everyone seems to do this – which annoys me intensely). The choice of model should be made a priori, NOT post-hoc. Choosing the model that looks better (or that better fits your preferences) is a major risk of bias. Study authors should look at their studies and how they are designed, the populations included, the doses, the time frames, the locations and anything else that is relevant and THEN decide what model is appropriate BEFORE doing the meta-analysis.
- Meta-analysis: you should at least briefly mention sensitivity analyses and why they are done (especially with known potential sources of heterogeneity).
- Meta-analysis: you should mention network meta-analyses, how they differ from pairwise meta-analyses, and what benefits they offer.
- Publication bias: you should mention here that funnel plots should not be undertaken with fewer than 10 studies even if the authors are not doing a regression test.
Minor points
- Line 70: “..a study find that…” should be “a study found that” (and the full stop is missing at the end of the sentence)
- Lines 137-139: this reads as though the risk is for the experienced, rather than inexperienced researches. You should rephrase this, for example instead of referring to the second approach, you should refer to the first approach thus: “The first approach is better suited to inexperienced systematic review authors, as it reduces the risk of producing an incomplete protocol…”
- Line 191: “…with disagreements resolved by a third reviewer.” should be “…with disagreements resolved by consensus or by a third reviewer”.
- Line 391: “…at the real beginning…” do you mean “…at the very beginning…”?
Author Response
Comment 1: De Cassai and coworkers have written a summary of best-practice methods for conducting a high-quality systematic review. Given the number of very poor-quality reviews that are published every year, such a guide would be useful. Generally speaking this article is pretty good. However, the manuscript requires some changes in order to make it reflect actual best practice. This should, however, be more than possible.
Answer 1: Thank you for your comments, we tried to improve our manucript following your suggestions as shown below.
Comment 2: Major points Your aim here was clearly to give a relatively brief overview of the most important aspects of doing a good quality SR. That is absolutely fine. However, as you are no doubt well aware, every one of your sections could be (and probably is) an entire book. Because it is not possible to cover all these details in your overview, you should provide the reader with a citation for gold standard, detailed instructions and guidance for each of the topics you cover.
Answer 2: We thank the reviewer for this valuable and insightful suggestion. We agree that each methodological domain discussed in our overview has a substantial body of detailed literature behind it. To improve the manuscript’s utility and academic rigor, we have now added references to authoritative, gold-standard guidance documents or resources in each relevant section (e.g., Cochrane Handbook, PRISMA 2020, JBI Manual), in addition to the key reference already incorporated (eg Swim guidlines, GRADE). These additions will enable readers to explore each topic in more depth and ensure alignment with current best practices.
Comment 3: Introduction: the gold standard for conducting systematic reviews of interventions is of course the Cochrane Collaboration. I understand that your manuscript considers systematic reviews of fields other than medicine and types other than interventions, but the Cochrane Handbook really is the bible for intervention reviews and should be discussed and cited.
Answer 3: We appreciate the reviewer’s recommendation. In recognition of its preeminence in guiding high‑quality intervention reviews, we have now explicitly discussed the Cochrane Collaboration and within the Introduction with the following paragraph
"Over recent decades, methodological standards for conducting SRs have become increasingly formalized, with the Cochrane Handbook for Systematic Reviews of Interventions widely regarded as the gold standard, particularly for intervention reviews in health care [3]. Although this manuscript addresses SRs across diverse fields and review types—including those beyond clinical interventions—the foundational methods outlined in the Cochrane Handbook remain highly influential and form the methodological bedrock for many review approaches."
Comment 4: Lines 70-71: I am not sure what you mean by “would have needed a major rewording of the research question”. You should expand a bit on this (one sentence). For example, perhaps you mean: “…would have needed a major rewording of the research question. That is, the findings of the study did not align with the research question”. Whatever it is that you mean, please add an explanation in your text.
Answer 4:vWe thank the reviewer for this helpful observation. We agree that the original sentence was unclear. We have now expanded the statement as follows:"A well-structured question serves as the foundation for the entire review process, guiding search strategies, study selection, and data synthesis. However, even if this is the most crucial step a study finds that 3 out of 10 studies would have needed a major rewording of the research question as the stated research questions were either too vague, inconsistent with the inclusion criteria, or misaligned with the actual objectives and findings "
Comment 5: Lines 72-76: you have forgotten the most comprehensive version: PICOTS – this includes the timeframe, which is vital for ensuring the data in the studies are comparable. I know some people include this in the Study design part, but in my experience, most authors do not remember to include the time component in their PICOS. Imagine you have an outcome like HbA1c, which is a measure of long-term glucose control, and then they measured the effects of an intervention after a day, and you combined this study with studies at 2 weeks and 6 months and 9 years. This would be inappropriate and inaccurate. Please include a short discussion of time in this section.
Answer 5:We appreciate the reviewer’s insightful suggestion. We agree that the timeframe (T) is a critical yet often overlooked component of structured research questions, particularly in studies involving outcomes where timing significantly influences measurement validity and comparability (e.g., HbA1c or even more survival!). We have revised the manuscript to include a brief discussion of PICOTS and the importance of specifying the time element in systematic review protocols as shown below:
"An even more comprehensive variation is PICOTS, even if rarely used, which adds Timeframe (T) to explicitly define the duration over which outcomes are assessed. This is particularly important for ecological or physiological outcomes that vary significantly over time. For instance, if assessing the impact of habitat restoration on bird population density, combining studies measuring outcomes after one breeding season with those measured after five years would likely yield misleading conclusions due to ecological lag effects."
Comment 6:Lines 79-96: these are actually not very comprehensive PICOS questions. They leave a lot of room for a study author to change the inclusion of studies to suit a predetermined preferred outcome. If I got a protocol to review with PICOS like this, I would send the authors back to fix them.
For the forest ecosystem question, it should be more like:
In degraded (defined as …) tropical forest ecosystems [anywhere they exist? Only in certain countries? What definition of tropical?], does reforestation (defined as…) with native tree species [what does “native” mean here? Native to the country, to the region, to that forest?], compared to natural recovery without intervention (defined as…), lead to greater increases species richness (defined as…) and abundance of native fauna [abundance = sheer numbers, increased numbers per species, something else? Also native meaning what here?] based on long-term [6 months, 1 year, 100 years? What is the time frame here?] observational studies [of any kind? Prospective only?]?
Similarly, the Pseudomonas question is not nearly detailed enough and should not be presented as a model for a good quality PICOS. The same is true for your example of a SPIDER question too.
Answer 6: we modified the PICOS in PICOTS as follow:
"
- In degraded tropical forest ecosystems, does reforestation with native tree species, compared to natural recovery without intervention, lead to greater increases in species richness and abundance of native fauna, based on long-term field observational studies?
Population (P): Degraded tropical forest ecosystems, defined as forests with >50% canopy loss, located within the Amazon Basin (including Brazil, Peru, Colombia), based on WWF ecoregion classifications.
Intervention(I): Reforestation efforts involving the planting of native tree species, defined as species naturally occurring in the respective ecoregion, excluding exotics and non-native cultivars.
Comparator(C): Natural recovery without active planting or management interventions, defined as passive regrowth following disturbance.
Outcome(O): Change in species richness (number of native plant and animal species per hectare, measured via standardized biodiversity surveys) and abundance of native fauna (total counts of individuals per species), assessed separately.
Timeframe(T): Minimum of 5 years post-intervention to capture long-term ecological recovery trajectories.
Study design(S): Prospective or retrospective observational studies with longitudinal monitoring, excluding short-term experiments (<1 year).
- In Pseudomonas aeruginosa cultures, does exposure to sub-lethal concentrations of ciprofloxacin, compared to no antibiotic treatment, reduce biofilm formation in in vitro experimental studies?
Population(P): Clinical isolates of Pseudomonas aeruginosa collected from hospitalized adult patients (>18 years) in tertiary care hospitals.
Intervention(I): Use of antibiotic A (e.g., ciprofloxacin) administered according to standard dosing guidelines.
Comparator(C): Use of antibiotic B (e.g., ceftazidime) or no antibiotic treatment (supportive care only).
Outcome(O): Antibiotic resistance development, defined as ≥4-fold increase in minimum inhibitory concentration (MIC) measured by broth microdilution assays.
Timeframe(T): Resistance assessed at baseline and after a minimum treatment period of 7 days.
Study design(S): Randomized controlled trials or observational cohort studies with appropriate resistance testing protocols."
and the spider example as follows:"
(3)How do adult members of Indigenous communities living adjacent to protected wildlife reserves in the Amazon Basin perceive the social and ecological impacts of community-based conservation programs, as explored through in-depth qualitative interviews? S (Sample): Adult members (≥18 years old) of Indigenous communities residing within 10 km of protected wildlife reserves in Brazil’s Amazon Basin (e.g., Kayapó, Yanomami territories); PI (Phenomenon of Interest): Perceptions and lived experiences related to community-based conservation programs, including views on biodiversity, land use, and cultural autonomy; D (Design): Qualitative studies using semi-structured or in-depth interviews, ethnographic methods, or participatory focus groups; E (Evaluation):
Thematic data describing perceived benefits (e.g., increased wildlife), concerns (e.g., reduced hunting access), and trust in conservation agencies;R (Research type):Qualitative studies only (excluding quantitative surveys or mixed-methods designs unless qualitative data are reported separately and in detail)."
Comment 7: Search strategy: you mention (lines 177-178) that clinical trial registries “are commonly employed” however, at least for reviews of interventions, this is absolutely essential. The Cochrane Collaboration considers that authors should at LEAST search the ICTRP. I have found on many occasions that data are available in, for example, ClinicalTrials.gov, that are not published and will never be published. This is frequently because the study did not find what the study sponsor would like. Neglecting to include clinical trials is a major risk of bias. You should change your manuscript to reflect this.
Answer 7: We modified "commonly employed" in "should be employed"
Comment 8: Study screening: you list all the sections of a PRISMA flow diagram (which is unnecessary because the reader can see this themselves), but you spend very little space talking about the purpose and details of screening.
You need to include a section on the creation of inclusion and exclusion criteria from the PICOTS, because it is NOT the same thing. You should also talk about the difference between screening abstracts (erring on the side of inclusion for example) and screening full texts (whittling down to truly relevant studies).
Answer 8: We modified the first part of the section as follows:"
Study screening is typically conducted in two stages: (1) title and abstract screening, and (2) full-text screening. At each stage, screening decisions should be based on predefined inclusion and exclusion criteria—which are developed from the review’s PICOTS (or equivalent) framework, but are not identical to it.
While PICOTS helps structure the research question, inclusion and exclusion criteria must translate that question into concrete, testable conditions. For example, if the “population” in PICOTS is “tropical forest ecosystems with >50% canopy loss,” the inclusion criteria must further specify what qualifies as “tropical,” and which geographic regions are eligible. Similarly, if the study design is limited to “observational studies with ≥5 years of follow-up,” the exclusion criteria must rule out short-term experiments, cross-sectional studies, or studies without clear follow-up durations.
During title and abstract screening, reviewers should generally err on the side of inclusion—retaining any study that potentially meets the criteria. This stage is intended to be sensitive, rather than specific. Full-text screening, in contrast, is more stringent and is used to exclude studies that do not fully meet the eligibility criteria. This step requires careful judgment and often necessitates pilot screening to calibrate reviewer decisions.
To minimize the risk of bias, this process should be performed independently by at least two reviewers, with disagreements resolved by a third reviewer. Emphasizing a double-screening approach is crucial, as evidence indicates that single-reviewer screening is associated with the omission of a significant number of eligible studies [17]."
Comment 9: "Heterogeneity (lines 284-291): although you write “approximately” when providing cut-offs for low moderate, and high heterogeneity, this is NOT what the Cochrane Handbook states. Heterogeneity is complex and nuanced, and the use of simple cutoffs is not appropriate. The Cochrane Handbook ACTUALLY says:
“Thresholds for the interpretation of the I2 statistic can be misleading, since the importance of inconsistency depends on several factors. A rough guide to interpretation in the context of meta-analyses of randomized trials is as follows:
-
- 0% to 40%: might not be important;
- 30% to 60%: may represent moderate heterogeneity*;
- 50% to 90%: may represent substantial heterogeneity*;
- 75% to 100%: considerable heterogeneity*.
*The importance of the observed value of I2 depends on (1) magnitude and direction of effects, and (2) strength of evidence for heterogeneity (e.g. P value from the Chi2 test, or a confidence interval for I2: uncertainty in the value of I2 is substantial when the number of studies is small).”
This is in section 10.10.2 of the current Cochrane Handbook."
Answer:9 Modified as follows:"I² values of approximately 0-40% might not be important, 30-60% may represent moderate heterogeneity, 50-90% may represent substantial heterogeneity, 75 -100% considerable heterogeneity. The importance of the observed value of I² depends on the magnitude and direction of effect and strength of evidence of heterogeneity (for example p-value from the chi square test), however the interpretation of heterogeneity is complex but important and should always be explored whenever possible (even through sensitivity analysis). It is important to remember that the uncertainty in the I² is substantial when the number of studies is small."
COmment 10: "Model choice: you nearly got this right. Your description of fixed vs random effects models is correct, but it is wrong to state that a random effects model is chosen as a consequence of substantial heterogeneity (even though everyone seems to do this – which annoys me intensely). The choice of model should be made a priori, NOT post-hoc. Choosing the model that looks better (or that better fits your preferences) is a major risk of bias. Study authors should look at their studies and how they are designed, the populations included, the doses, the time frames, the locations and anything else that is relevant and THEN decide what model is appropriate BEFORE doing the meta-analysis."
Answer 10 :"The choice between a fixed or a random effect model should be performed a priori; a fixed effects models should be preferred over random effect models only if researchers are absolutely certain that all the studies included do not have confounding effects or heterogeneity, a situation that rarely takes place."
Comment 11: Meta-analysis: you should at least briefly mention sensitivity analyses and why they are done (especially with known potential sources of heterogeneity).
Answer 11: Added the following:"
"When potential sources of heterogeneity are known or suspected—such as differences in populations, interventions, or study quality—sensitivity analyses should be conducted. These involve re-running the meta-analysis after excluding specific subsets of studies (e.g., low-quality studies, studies with extreme effect sizes) to assess how robust the overall findings are and they are helpful to potentially identify the source of heterogeneity."
Comment 12: "Meta-analysis: you should mention network meta-analyses, how they differ from pairwise meta-analyses, and what benefits they offer."
Answer 12: We agree with the Reviewer that NMA are of potential interest, but they would need a completely different paper, however, we mentioned it at the end of the meta-analysis paragraph as follows:"In addition to traditional pairwise meta-analyses, which compare two interventions at a time, network meta-analyses (NMA) allow for simultaneous comparisons of multiple interventions, even if some have not been directly compared in any individual study. NMA integrates both direct and indirect evidence using a common comparator (e.g., placebo), enabling the ranking of interventions in terms of effectiveness. This approach is particularly valuable in fields where multiple treatment or management strategies exist, and head-to-head trials are sparse."
Comment 13:"Publication bias: you should mention here that funnel plots should not be undertaken with fewer than 10 studies even if the authors are not doing a regression test."
Answer 13: Modified as follow:"
When there are a sufficient number of studies (usually ten or ore) publication bias can be evaluated with funnel plots through visual inspection for asymmetry, potentially suggesting the presence of publication bias or other small-study effects. However, visual interpretation can be subjective and inconsistent. Therefore, statistical tests such as Egger’s regression test can be employed to provide a more objective and reproducible assessment of funnel plot asymmetry [28]."
Minor points
Comment 13: Line 70: “..a study find that…” should be “a study found that” (and the full stop is missing at the end of the sentence)
Answer 13: Taken
Comment 14:Lines 137-139: this reads as though the risk is for the experienced, rather than inexperienced researches. You should rephrase this, for example instead of referring to the second approach, you should refer to the first approach thus: “The first approach is better suited to inexperienced systematic review authors, as it reduces the risk of producing an incomplete protocol…”
Answer 14:Taken
Comment 15: "Line 191: “…with disagreements resolved by a third reviewer.” should be “…with disagreements resolved by consensus or by a third reviewer”."
Answer 15 :Taken
Comment 16:"Line 391: “…at the real beginning…” do you mean “…at the very beginning…”?"
Answer 16: Taken
Reviewer 3 Report
Comments and Suggestions for Authors
The manuscript "Methodological Standards for Conducting High-Quality Systematic Reviews" submitted to Biology journal by MDPI presents a well-structured and comprehensive guide to conducting systematic reviews. Please find my comments for improvement below:
1. The title "Methodological Standards for Conducting High-Quality Systematic Reviews" is accurate, concise, and directly reflects the content and aim of the manuscript. However, it does not specifically address the biological context suggested by the journal’s scope, which could limit its visibility in biology-specific searches.
2. The abstract is clear and well-aligned with the main body. It succinctly summarizes the purpose, steps involved, and significance of systematic reviews, mentioning the critical guidelines (GRADE, PRISMA) which are relevant and widely accepted in the scientific community. However, it states explicitly "health and social sciences," whereas the scope should explicitly include biology to fit the journal's readership.
3. The introduction effectively frames the rationale for the manuscript by highlighting the value of systematic reviews in synthesizing research. It accurately outlines the current challenges in systematic reviews and emphasizes the importance of quality guidelines. It provides justification but could benefit from more robust epidemiological data or statistics illustrating the impact or frequency of poor-quality systematic reviews, particularly in biology. The novelty is limited to a comprehensive explanation of standard practices. However, framing explicitly around biology (as suggested) would substantially increase its relevance to the journal.
4. The manuscript elaborates clearly on general findings, integrates guidelines like GRADE, and adequately compares these methods with international best practices (e.g., Cochrane Handbook, SWiM). There is scope to enrich the discussion by integrating a brief comparison with other systematic review methodologies, such as RAMESES or meta-aggregation.
5.The PRISMA diagram provided is illustrative; however, it omits a box for "records from previous systematic reviews," This inclusion would better reflect current best practices.
6. Appropriately covers GRADE for assessing the certainty of evidence in systematic reviews. However, a reference to appraisal tools for *in vitro* studies, such as those discussed in Iqhrammullah et al. Sci. Pharm. (2023), is missing, as recommended. Inclusion of such references is crucial for systematic reviews conducted in biological and pharmaceutical sciences.
7. The manuscript would strongly benefit from explicitly recommending biology-specific databases such as CABI or BIOSIS, enhancing its utility to biology researchers as you suggested.
8. However, the conclusion currently lacks explicit framing within biology, somewhat misaligning with the journal's focus. A final emphasis on how these guidelines specifically advance systematic reviews in biological research is recommended.
9. The scholarly content provided is comprehensive, detailed, and highly informative, suitable for major publishers including MDPI. However, considering its targeted journal (Biology), the manuscript needs specific refinements to emphasize its relevance to biology. Without these biology-focused adjustments, rejection or recommendation for transfer to another journal with a more general methodological scope, such as *Publications* (also MDPI), could be justified.
Author Response
Comment 1. The title "Methodological Standards for Conducting High-Quality Systematic Reviews" is accurate, concise, and directly reflects the content and aim of the manuscript. However, it does not specifically address the biological context suggested by the journal’s scope, which could limit its visibility in biology-specific searches.
Answer 1: We would like to thank the Reviewer for his comment. If possible we would like to keep the title as it stands. While it is true the the manuscript is submitted to Biology (and most of the example and content is based for biology readers) it is may catch the attention also of other readers and for this reason it could be helpful to keep the title for a broader audience (however if the editor and reviewer believe it is strictly necessary we will modify it)
Comment 2. The abstract is clear and well-aligned with the main body. It succinctly summarizes the purpose, steps involved, and significance of systematic reviews, mentioning the critical guidelines (GRADE, PRISMA) which are relevant and widely accepted in the scientific community. However, it states explicitly "health and social sciences," whereas the scope should explicitly include biology to fit the journal's readership.
Answer 2: Modified accordingly ". This article offers a detailed guide to conducting high-quality systematic reviews in biology, health and social sciences"
Comment 3. The introduction effectively frames the rationale for the manuscript by highlighting the value of systematic reviews in synthesizing research. It accurately outlines the current challenges in systematic reviews and emphasizes the importance of quality guidelines. It provides justification but could benefit from more robust epidemiological data or statistics illustrating the impact or frequency of poor-quality systematic reviews, particularly in biology. The novelty is limited to a comprehensive explanation of standard practices. However, framing explicitly around biology (as suggested) would substantially increase its relevance to the journal.
Answer 3: We have added the following paragraph:"In a survey of 102 SRs and meta-analyses in ecology and evolutionary biology (2010–2019), only ~16% referenced any reporting guideline, and those that did scored significantly higher on reporting quality metrics than the average [3]. Moreover, reviews dating back to the early 2000s highlighted that ecological meta-analyses often lacked essential methodological rigour—such as protocol registration, risk of bias assessments, or clear inclusion criteria—revealing a gap relative to established practices in clinical research [4] "
Comment 4. The manuscript elaborates clearly on general findings, integrates guidelines like GRADE, and adequately compares these methods with international best practices (e.g., Cochrane Handbook, SWiM). There is scope to enrich the discussion by integrating a brief comparison with other systematic review methodologies, such as RAMESES or meta-aggregation.
Answer 4:We appreciate this thoughtful suggestion. In response, we have added a short section in the discussion comparing our focus on widely used approaches (e.g., Cochrane, GRADE, SWiM) with alternative methodologies such as RAMESES for realist reviews and meta-aggregation for qualitative synthesis. This addition clarifies the scope of our manuscript and highlights the existence of other robust review methods suited to different research aims and evidence types.
"
Alternative Approaches to Evidence Synthesis
While this manuscript focuses on broadly applicable, quantitative synthesis approaches (e.g., GRADE, SWiM, Cochrane Handbook), it is important to acknowledge alternative methodologies developed for different types of evidence and research questions. For instance, the RAMESES (Realist And MEta-narrative Evidence Syntheses: Evolving Standards) [46] framework supports realist reviews, which aim to understand how and why interventions work (or fail) in particular contexts—especially in complex, system-level settings. Similarly, in qualitative research, meta-aggregation (as used in the Joanna Briggs Institute model) [47] offers a structured approach for synthesizing findings while preserving participants’ original meaning. Unlike thematic synthesis, meta-aggregation emphasizes transparency and alignment with primary study conclusions. These alternative methodologies serve distinct purposes and are complementary to the general frameworks described here. Researchers should carefully consider their review goals and the nature of the available evidence when selecting an appropriate synthesis method.
"
Comment 5.The PRISMA diagram provided is illustrative; however, it omits a box for "records from previous systematic reviews," This inclusion would better reflect current best practices.
Answer 5: While this is an interesting point the box "from previous systematic reviews" is actually not shown even in the PRISMA "official" flowchart. For this reason we would avoind inserting it
Comment 6. Appropriately covers GRADE for assessing the certainty of evidence in systematic reviews. However, a reference to appraisal tools for *in vitro* studies, such as those discussed in Iqhrammullah et al. Sci. Pharm. (2023), is missing, as recommended. Inclusion of such references is crucial for systematic reviews conducted in biological and pharmaceutical sciences.
Answer 6: We ask Editor and Reviewer guidance in appropriately repond to this concern as we do not understand how to cite the following paper in our manuscript "Iqhrammullah, M.; Rizki, D.R.; Purnama, A.; Duta, T.F.; Harapan, H.; Idroes, R.; Ginting, B. Antiviral Molecular Targets of Essential Oils against SARS-CoV-2: A Systematic Review. Sci. Pharm. 2023, 91, 15. https://doi.org/10.3390/scipharm91010015"
Moreover, we would kindly ask to provide the correct doi if we have mistaken the paper.
Comment 7. The manuscript would strongly benefit from explicitly recommending biology-specific databases such as CABI or BIOSIS, enhancing its utility to biology researchers as you suggested.
Answer 7: Taken
Comment 8. However, the conclusion currently lacks explicit framing within biology, somewhat misaligning with the journal's focus. A final emphasis on how these guidelines specifically advance systematic reviews in biological research is recommended.
Answer 8 :We thank the reviewer for this insightful suggestion. In response, we have revised the conclusion to explicitly frame the discussion within the context of biological research. We now emphasize how the methodological guidance provided throughout the manuscript addresses common challenges in biological systematic reviews and aims to raise standards of transparency, reproducibility, and rigor in this field.
"In summary, this guide aims to consolidate widely accepted standards for conducting systematic reviews and adapt them to the specific needs of researchers in the biological sciences. From diverse study designs and outcome measures to field-specific challenges like ecological variability and non-standardized interventions, systematic reviews in biology require careful methodological attention. By contextualizing established frameworks—such as PRISMA, GRADE, and SWiM—and highlighting additional tools relevant to biology (e.g. SPIDER), this manuscript supports researchers in producing high-quality, transparent, and reproducible evidence syntheses. Strengthening the methodological foundation of systematic reviews in biology is essential for advancing reliable, evidence-based decision-making in conservation, ecology, agriculture, and biomedical science."
Comment 9. The scholarly content provided is comprehensive, detailed, and highly informative, suitable for major publishers including MDPI. However, considering its targeted journal (Biology), the manuscript needs specific refinements to emphasize its relevance to biology. Without these biology-focused adjustments, rejection or recommendation for transfer to another journal with a more general methodological scope, such as *Publications* (also MDPI), could be justified.
Answer 9: We would like the Reviewer for the time he spent reviewing our manuscript and providing guidance in further improve the overall quality of our manuscript
Reviewer 4 Report
Comments and Suggestions for Authors
In my opinion, I was waiting such a mansucript as a guideline for properly preparing systematic reviews. This article offers a detailed guide to conducting high-quality systematic reviews in health and social sciences. The authors of the manuscript are also presenting the relative methods and tools for synthesizing data, covering both quantitative approaches like meta-analysis and qualitative narrative synthesis when statistical pooling is not possible. Anyway, there are some issues, which I have found that needs improvements, like the abstract where the authors used ";", instead of ",". In addition, I would recommend them to improve the quality of the scheme presented in the Figure 1, while regarding the mentioned tools in the Table 1, I suggest them to include the relative citation and reference for each of the tools.
These are the issues, which need to be addressed by the authors in order to have a good manuscript for being published in the journal.
Author Response
We thank the reviewer for his comments, we corrected some typos all over the manuscript and we have provided the relevant reference list for each mentioned tool
Round 2
Reviewer 3 Report
Comments and Suggestions for Authors
I still feel the manuscript is disconnected from the essence of biological inquiry. The language and structure lean heavily toward general evidence-based medicine and health sciences methodology. Authors may opt to do the followings:
1. Anchor examples in Biological Sciences (such as Ecology, Genetics, Molecular Biology). Most of the practical illustrations focus on medical or health-related interventions (e.g., Pseudomonas exposure to ciprofloxacin).I suggest to add more domain-specific examples from subfields like ecology, conservation biology, evolutionary biology, microbiology, or molecular biology. For instance, instead of just antibiotic resistance, you could explore: “In Arabidopsis thaliana, how does exposure to salicylic acid influence gene expression compared to control conditions?"
2. The review doesn’t sufficiently address the unique methodological problems in biological research (e.g., variability in ecosystems, species-specific outcomes, lab-vs-field inconsistencies). Authors should add subsection(s) that explicitly discusses: (a) Non-standardized outcome measures in ecological and evolutionary studies. (b) Species-specific or strain-specific biological responses. (c) High heterogeneity due to environmental contexts in biodiversity or phylogenetic research.
3. While PRISMA, GRADE, and Cochrane tools are standard in medicine, biology researchers also use domain-specific tools and repositories. Please explore tools like SYRCLE’s risk of bias tool (already included—good); databases like GenBank, BOLD Systems, or GBIF for sourcing biological data. Repositories such as Dryad, BioStudies, or Zenodo relevant to biology.
4. The title and framing are quite general and may not feel owned by biological researchers. I think authors may consider reframing the title(?)
5. In the Abstract and Introduction, explicitly state how the paper aims to translate evidence-based review principles into the biological sciences context.
6. Unfortunately, there’s no illustration from published systematic reviews in biology (e.g., meta-analyses on species richness, CRISPR editing efficiency, or population genetics). Filling these gaps can help to direct the article into biology-relevant paper.
7. Consider to change the article type into Opinion or other types. Review article type seems not appropriate because it supposedly synthesized data from published study. Although the manuscript is labeled as a review article, it does not synthesize or appraise primary studies. Instead, it provides methodological recommendations and guidance. Therefore, I recommend reclassifying this as an Opinion, Perspective, or Methodology article, depending on the Biology’s available categories.
Author Response
Comment 0: still feel the manuscript is disconnected from the essence of biological inquiry. The language and structure lean heavily toward general evidence-based medicine and health sciences methodology. Authors may opt to do the followings:
Answer0: We are really grateful for the quick response by the Reviewer as with his guidance we believe our manuscript is now better suited for the readers.
Comment 1. Anchor examples in Biological Sciences (such as Ecology, Genetics, Molecular Biology). Most of the practical illustrations focus on medical or health-related interventions (e.g., Pseudomonas exposure to ciprofloxacin).I suggest to add more domain-specific examples from subfields like ecology, conservation biology, evolutionary biology, microbiology, or molecular biology. For instance, instead of just antibiotic resistance, you could explore: “In Arabidopsis thaliana, how does exposure to salicylic acid influence gene expression compared to control conditions?"
Answer 1: We thank the reviewer for this excellent suggestion. We agree that including more examples from core biological subfields will enhance the manuscript’s relevance and appeal to a broader scientific audience beyond clinical or health-related domains.
In response, we have incorporated two new domain-specific examples:
-
In the section on formulating research questions using the PICOTS framework, we added an example from molecular biology exploring transcriptional responses in Arabidopsis thaliana following salicylic acid exposure.
-
In the SPIDER framework section, we added a new example focused on early-career researchers in molecular biology and their perceptions of CRISPR-Cas9 gene editing technologies, derived from qualitative interview-based research.
These examples are now presented alongside the existing ones
Comment 2. The review doesn’t sufficiently address the unique methodological problems in biological research (e.g., variability in ecosystems, species-specific outcomes, lab-vs-field inconsistencies). Authors should add subsection(s) that explicitly discusses: (a) Non-standardized outcome measures in ecological and evolutionary studies. (b) Species-specific or strain-specific biological responses. (c) High heterogeneity due to environmental contexts in biodiversity or phylogenetic research.
Answer 2: We agree with the Reviewer. We have added the following subsection to briefly explain unique challenge in biology meta-analysis.
"
- Unique Methodological Challenges in Biological Research
Biological research, particularly in ecology, evolutionary biology, and related subfields, faces several methodological challenges that are distinct from clinical or health-related studies. These challenges stem from the complexity of living systems and their interactions with highly variable environments.
(a) Non-standardized outcome measures in ecological and evolutionary studies
Unlike clinical trials with well-established and standardized endpoints (e.g., blood pressure or mortality), ecological and evolutionary studies often rely on diverse outcome measures such as species richness, biomass, gene expression levels, or behavioral traits. This diversity can make it difficult to directly compare results across studies or synthesize findings through meta-analyses [48]. Moreover, these outcomes may be measured using different methodologies, scales, or temporal resolutions, increasing methodological heterogeneity and complicating systematic reviews [49].
(b) Species-specific or strain-specific biological responses
Variation between species, strains, or populations is a hallmark of biological research, reflecting differences in genetics, physiology, life history, and ecological niches. Such biological heterogeneity means that responses to experimental treatments or environmental factors can differ markedly, even within seemingly similar contexts. For example, a pesticide might have a strong toxic effect on one insect species but be benign to another. Failing to account for these differences risks overgeneralizing results and undermines the validity of pooled analyses [50]. Researchers must carefully consider taxonomic and genetic diversity when designing studies and synthesizing data.
(c) High heterogeneity due to environmental contexts in biodiversity or phylogenetic research
Environmental variability introduces another significant source of heterogeneity in biological research. Spatial factors (e.g., latitude, habitat type), temporal dynamics (e.g., seasonal variation), and anthropogenic impacts (e.g., pollution, land use changes) all affect biological outcomes and can lead to conflicting or context-dependent findings. Biodiversity studies, in particular, must grapple with this complexity when synthesizing data from multiple ecosystems or geographic regions. Phylogenetic relationships further complicate analyses, as closely related species may share traits that influence responses, requiring sophisticated statistical models to control for evolutionary relatedness [51,52]. These challenges highlight the need for explicit modeling of heterogeneity and sensitivity analyses in biological meta-research."
Comment 3. While PRISMA, GRADE, and Cochrane tools are standard in medicine, biology researchers also use domain-specific tools and repositories. Please explore tools like SYRCLE’s risk of bias tool (already included—good); databases like GenBank, BOLD Systems, or GBIF for sourcing biological data. Repositories such as Dryad, BioStudies, or Zenodo relevant to biology.
Answer 3: We would like to thank the reviewer. The paragraph now reads:"
The Cochrane Handbook for Systematic Reviews of Interventions [5] advises searching multiple databases to minimize the risk of missing relevant studies. Commonly used databases include MEDLINE/PubMed, EMBASE, Scopus, Cochrane Central Register of Controlled Trials (CENTRAL), CINAHL, PsycINFO, CABI and BIOSIS. However, this list is not exhaustive; researchers are encouraged to expand their search to as many relevant databases as resources allow. In addition to these general databases, it is essential to highlight biology domain-specific resources such as GenBank for genetic sequences, BOLD Systems for DNA barcoding data, and GBIF for biodiversity records. Furthermore, repositories like Dryad, BioStudies, and Zenodo facilitate open data sharing and promote transparency and reproducibility in biological research. Utilizing these resources helps researchers access diverse datasets essential for addressing complex biological questions across molecular biology, ecology, and evolutionary studies."
Comment 4. The title and framing are quite general and may not feel owned by biological researchers. I think authors may consider reframing the title(?)
Answer 4: We agree that the title is general, however, as other reader could be interested to read our paper (and not only in the biology setting) we believe we ask to leave the title as it is (however if the Reviewer and Editor feel absolutely necessary to change it we will do it)
Comment 5. In the Abstract and Introduction, explicitly state how the paper aims to translate evidence-based review principles into the biological sciences context.
Answer 5: Added the following at the end of the abstract :"This paper aims to adapt and translate evidence-based review principles, commonly applied in clinical research, into the context of biological sciences. By highlighting domain-specific methodologies, challenges, and resources, we provide tailored guidance for researchers in ecology, molecular biology, evolutionary biology, and related fields to conduct transparent, and reproducible evidence syntheses." and the following in the introduction "
This paper seeks to translate these principles into the biological context by addressing unique methodological challenges and incorporating domain-specific tools and resources. The aim of this paper is to offer a comprehensive overview of how to conduct a high-quality systematic review, with detailed discussions of each step based on internationally recognized guidelines and best practices."
Comment 6. Unfortunately, there’s no illustration from published systematic reviews in biology (e.g., meta-analyses on species richness, CRISPR editing efficiency, or population genetics). Filling these gaps can help to direct the article into biology-relevant paper.
Answer 6: We appreciate the reviewer’s suggestion to include illustrative images from published systematic reviews in biology. While we have chosen not to include specific figures from published papers to avoid copyright concerns and to focus on conceptual clarity, we are open to any further suggestions the reviewer might have regarding alternative ways to enhance the manuscript visually.
Comment 7. Consider to change the article type into Opinion or other types. Review article type seems not appropriate because it supposedly synthesized data from published study. Although the manuscript is labeled as a review article, it does not synthesize or appraise primary studies. Instead, it provides methodological recommendations and guidance. Therefore, I recommend reclassifying this as an Opinion, Perspective, or Methodology article, depending on the Biology’s available categories.
Answer 7: We thank the reviewer for this thoughtful recommendation regarding the article type. We would be glad to consider reclassifying the manuscript as an Opinion, Perspective, or Methodology article if the journal allows it. However, to our understanding, the journal primarily publishes Original Research and Review articles. That said, if the editor believes reclassification is possible and appropriate, we are open and happy to proceed accordingly.